# Urinary Bladder Schistosomiasis Mimicking Neoplasm: A Case Report

**DOI:** 10.3390/medicina58081001

**Published:** 2022-07-27

**Authors:** Majid Darraj

**Affiliations:** Department of Medicine, College of Medicine, Jazan University, Jazan 45142, Saudi Arabia; mdarraj@jazanu.edu.sa

**Keywords:** schistosomiasis, urinary bladder, neoplasm, praziquantel

## Abstract

Schistosomiasis is a neglected disease that is prevalent in tropical and subtropical areas. A 20-year-old woman presented to the emergency room with a history of right flank pain and lower abdominal discomfort for one day, which coincided with the onset of menses. The patient did not provide any history of premenstrual hematuria. The physical examination revealed right costovertebral angle tenderness and was otherwise unremarkable. The urinalysis demonstrated a mild increase in red and white blood cells and no ova or parasite. The blood test was normal, except for eosinophilia. A right pedunculated intraluminal urinary bladder mass was detected by the computerized axial tomographic scan and ultrasonography, and after the transurethral resection of the mass, the patient was diagnosed with urinary schistosomiasis. The patient received two doses of oral praziquantel of 1200 mg every 12 h for one day. The cure was confirmed with a one-month post-treatment follow-up that revealed a normal urine microscope and eosinophil count. The *S. haematobium* infection should be evaluated as a possible cause of urinary bladder lesion in those who have travelled or lived in endemic areas.

## 1. Introduction

Schistosomiasis is a parasitic infection that is caused by blood flukes (trematode worms) of the Schistosoma genus in tropical and subtropical areas. According to the statistics, approximately 236.6 million people required preventative care in 2019, 90% of whom were from Africa (WHO, 2022), with approximately 41,000 deaths annually [1,2,3,4,5,6,7,8,9]. Morbidity will be reduced and prevented with treatment, and 51 endemic nations with moderate-to-high transmissions require preventative chemotherapy for schistosomiasis, in which persons and communities are targeted for large-scale treatment. The strategic plan of the World Health Organization (WHO) for schistosomiasis for 2012–2020 aims to eliminate schistosomiasis as a public health issue by 2020, decrease morbidity by 2020, and prevent transmission in some places by 2025 [9,10]. Infection occurs by direct contact with fresh water that is contaminated with cercariae, which when released from their snail host, penetrate the skin of the human host. They then migrate through the systemic circulation to their final destination and ultimately reside in various venous plexi that are somewhat species specific. *S. haematobium* usually resides in the venous plexus of the urinary bladder and rarely in the rectal venules, where the female adult worms deposit eggs, which gradually migrate toward the bladder mucosa and ureters and are shed in the urine. The eggs lodge in the bladder wall, which leads to ulceration and a local granulomatous inflammatory response [3]. The prevalence of shistosomiasis in Saudi Arabia is not common, although the country has an endemic history with schistomiasis countries, such as Egypt, Yemen, and Iraq [11]. The regions whereby schistosomiasis may be identified are progressively expanding in Europe as a result of the recent increase in the northward migration movement from Africa through the Mediterranean area with foci in Spain, Portugal, Sicily, and Greece [12].

## 2. Case Presentation

An otherwise healthy 20-year-old Ethiopian immigrant woman presented to the emergency department with a one-day history of right flank pain and lower abdominal discomfort, which coincided with the onset of menses. She did not have a history of premenstrual hematuria. The physical examination revealed tenderness in the right costovertebral angle. The abdomen was soft with no tenderness. Additionally, there was a normal bowel sound and no added sounds upon auscultation of the abdomen. The other systems were unremarkable.

The urinalysis demonstrated a mild increase in the 21–50 red blood cells (20–50/HPF) and white blood cells (11–20 white blood cells/HPF). The urine cytology did not reveal any tumor cells, and the microscopic examination of the urine did not demonstrate any ova or parasites. Nitrites were not observed, and no growth was observed in the urine culture. The complete blood count revealed a white blood cell count of 5.7 × 10^9^/L (range 4.5–11 × 10^9^/L), hemoglobin 132 g/L (range 120–160 g/L), and platelets 381 × 10^9^/L (range 140–440 × 10^9^/L). The differential white blood cell count was normal, except for mild eosinophilia, with 1.15 × 10^9^/L (range 0.0–0.4 × 10^9^/L), or 18.0% (range 0.0–5.0%). The blood chemistry levels were all within normal ranges. A computerized axial tomographic scan (CT) of the abdomen and pelvis was performed and showed no pathologically enlarged lymph nodes. However, it revealed a 1.5 by 1.5 by 0.9 cm pedunculated intraluminal urinary bladder mass on the right side. The interpreting radiologist suggested that the mass could be better assessed by an ultrasound, which is shown in Figure 1. Although the ultrasound showed no stone, it raised concerns regarding possible bladder neoplasia. The patient underwent a cystoscopy with a transurethral resection of the mass for a histological assessment. The microscopic examination, as shown in Figure 2, revealed organisms that were morphologically compatible with the eggs of *S. haematobium*. We prescribed the patient 1200 mg of praziquantel orally every 12 h for one day (two doses). The one-month post-treatment follow-up included monitoring for eosinophilia and a microscopic examination of the urine (*S. haematobium*), which revealed a normal reading that confirmed that the patient had been cured.

## 3. Discussion

Schistosomiasis is a tropical disease that is caused by the following six different species: *S. haematobium, S. mansoni, S. japonicum, S. mekongi, S. intercalatum*, and *S. guineensis. While S. haematobium* is responsible for urinary schistosomiasis, the other species cause intestinal schistosomiasis [1,2].

Most cases occur in sub-Saharan Africa [1,2]. The clinical manifestation of schistosomiasis can be divided into acute and chronic. Acute schistosomiasis (Katayama syndrome), which is a serum sickness-like illness that presents with fever, headaches, urticaria, abdominal pain, and lymphadenopathy, occurs particularly in the non-immune host approximately two to 12 weeks after the significant first exposure to larvae prior to the development of adult worms and egg deposition [4]. Therefore, praziquantel treatment is not indicated for acute schistosomiasis, as there are not yet any adult worms present to eradicate. The chronic disease is due to a granulomatous inflammatory response against the egg antigens in the host tissue [5]. Treatment of the chronic disease includes using praziquantel to eradicate the adult worms and cease the production of new eggs. Most of the urinary diseases that are caused by schistosomes gradually regress; however, 10% of those who are infected will develop long-standing complications, such as chronic bladder ulceration, bladder contracture, stricture of the ureters, renal failure, and, ultimately, bladder carcinoma [6]. Schistosomiasis should be considered in persons who have a history of freshwater exposure in endemic areas. *Schistosoma* spp. infections can be asymptomatic and lack clinical findings and eosinophilia [4].

Our patient presented with symptoms that were unrelated to her bladder lesion. The subsequent investigation of the flank pain revealed the incidental finding of the bladder mass, which was resected to reveal manifestations of *S. haematobium*. However, the patient had moderate eosinophilia, which suggested the presence of a helminthic infection such as schistosomiasis.

Schistosomiasis can be diagnosed by a microscopic examination of the urine (*S. haematobium*) or stool (*S. mansoni, S. japonicum*), where ova may be observed. However, with our patient, ova were not observed in the urine. A stool analysis was not performed at the time of presentation or initiation of therapy. Specimen collection is ideally performed at least two months after the last exposure to account for the time that is needed for the adult worms to develop and produce eggs. Repeated stool or urine examinations are recommended, as the passage of ova can be in small numbers or intermittent. Serologic assays for schistosoma spp. antibodies are available and can be detected earlier than in a microscopic examination for eggs. However, serology assays do not distinguish between active and prior infections and remain positive for life, which renders serology diagnostically less useful for patients with a history of treated schistosomiasis. Serology, however, is useful in non-immune individuals (returned travelers) with more recent potential exposure. Additionally, a polymerase chain-reaction technique detection of schistosome DNA in feces or blood has been developed, which can provide a diagnosis at any stage of infection, particularly in early infections, where antibodies might not yet have developed and eggs may not yet have been produced [7].

As bladder schistosomiasis predisposes individuals to bladder squamous cell carcinomas, CT scans, magnetic resonance imaging, and ultrasonography are sensitive modalities for the evaluation of urinary tract complications that are related to schistosomiasis. The WHO recommends ultrasounds as the preferred modality in endemic countries with schistosomiasis (the Niamey-Belo Horizonte Protocol) [1,8,13]. Praziquantel (a pyrazinoisoquinoline derivative) is the drug of choice for treating all species of schistosomes and can be used by pregnant and lactating women. The side effects are usually mild and include nausea, urticaria, and dizziness, which are thought to be related to the death of the parasites rather than the drug itself. The recommended dose for *S. haematobium* is 40 mg/kg per day orally in two divided doses that are taken 12 h apart. Although it is well absorbed, it has extensive first-pass hepatic clearance. A single treatment has an efficacy of 66–95%, with the second treatment four to six weeks later, increasing the efficacy to 95–100%. The second dose is recommended if eosinophilia persists (7). The post-treatment follow-up includes monitoring for eosinophilia and a microscopic examination of the urine *(S. haematobium*) or stool (*S. mansoni, S. japonicum*) one to two months post-treatment to confirm that it has been cured [3].

## 4. Conclusions

In conclusion, this case report presents a 20-year-old woman who presented to the emergency room with a history of right flank pain and lower abdominal discomfort for one day, which coincided with the onset of menses. The patient was diagnosed with urinary schistosomiasis and received two doses of oral praziquantel of 1200 mg every 12 h for one day. The cure was confirmed with a one-month post-treatment follow-up that revealed a normal urine microscope and eosinophil count. As an effective drug, Praziquantel is recommended to treat all forms of schistosomiasis.

## Figures and Tables

**Figure 1 medicina-58-01001-f001:**
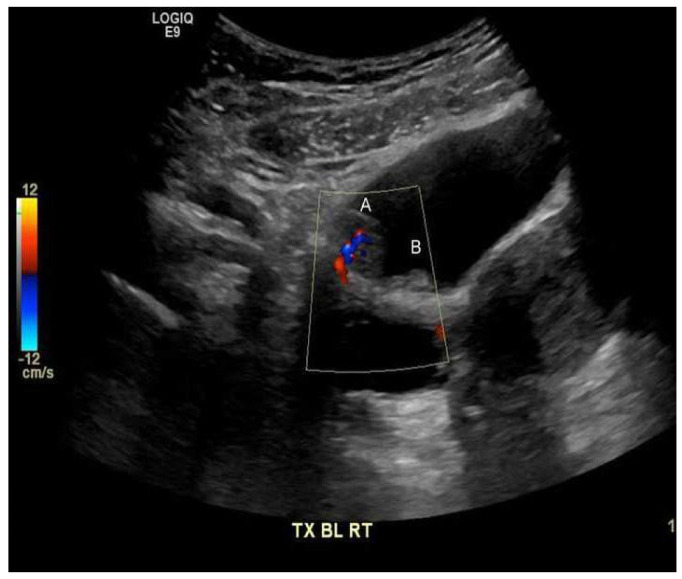
Polypoid lesion along the right anterior superior wall of the urinary bladder (**A**) 1.5 by 1.5 by 0.9 cm, with the internal flow on the color Doppler imaging suggesting that the mass has a blood flow. Medially, a smaller adjacent polypoid lesion (**B**) measuring 0.9 cm is noted.

**Figure 2 medicina-58-01001-f002:**
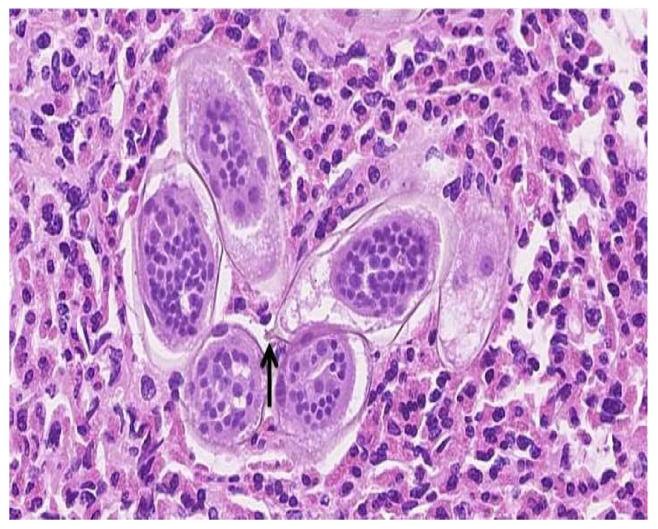
High magnification (approximately 40×) of a hematoxylin and eosin-stained section of the large bladder lesion resected transurethrally. Numerous elongated *Schistosoma* spp. ova with viable miracidia in the background of many eosinophils are identified. A terminal spine (arrow) is present on one of the eggs, which confirms that *S. haematobium* is the responsible pathogen.

## Data Availability

All data shown in this study are included in this published article.

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
