# Peer review of "Urinary Bladder Schistosomiasis Mimicking Neoplasm: A Case Report"

_medicina, 2022, doi:10.3390/medicina58081001_

Round 1

Reviewer 1 Report

Majid A. Darraj described a case of urinary schistosomiasis in Saudi Arabia.

-The case report is quite well described.

-I would express the use according to guidelines, sequential ultrasound study and then subsequently tomographic.

-I would add a prevalence of this infection also in Western populations.

- I advise the author to comment more incisively on the particularity of the clinical case.

- unify the comments of each figure legend relating to Figure 1 or Figure 2.

- insert the periods at the end of the sentence of Figure 1 and Figure 2.

Author Response

Thank you for your valuable comments

-I would express the use according to guidelines, sequential ultrasound study and then subsequently tomographic.

Absolutely, However the CT in this case was done prior US, and thus we highlight that both modalities are quite sensetive 

-I would add a prevalence of this infection also in Western populations.

thank you, the comment was considered and 2 references were added.

- I advise the author to comment more incisively on the particularity of the clinical case.

the comment was considered and done.

- unify the comments of each figure legend relating to Figure 1 or Figure 2.

done

- insert the periods at the end of the sentence of Figure 1 and Figure 2.

done

Reviewer 2 Report

This is a simple case report of urinary schistosomiasis in an Ethiopian immigrant in Saudi Arabia. The case is well described but fairly typical of such presentations, and thus well written but not perhaps very novel.

Author Response

Thank you for your feedback

Reviewer 3 Report

The authors of “Urinary Bladder Schistosomiasis Mimicking Neoplasm: A Case Report” present a very interesting case report on a 20 year old woman with urinary bladder schitosomiasis. They do an excellent job of presenting the clinical case as well as an extensive review of the literature for this condition. I only have minor edit suggestions and that is that the genus and species of the Schistosoma organism should be italicized. Here are the line numbers in the document where this should occur:

Line 22, Line 38, Line 69, Line 72, Lines 95-96, Line 116, Line 119-120, Line 150

Author Response

Thank you for your feedback

Schistosoma organism should be italicized. Here are the line numbers in the document where this should occur

done 
